

# The ability of laying pullets to negotiate two ramp designs as measured by bird preference and behaviour

Isabelle C. Pettersson, Claire A. Weeks, Kate I. Norman and Christine J. Nicol

Bristol Veterinary School, University of Bristol, Bristol, United Kingdom

## ABSTRACT

**Background**. Laying hens are often kept in barn or free-range systems where they must negotiate level changes in the house to access resources. However, collisions and resultant keel fractures are commonplace. Producers sometimes add ramps to make raised areas more accessible but designs vary and very little research has investigated bird preference or behaviour when using different ramp designs, or the effect of ramp design on falls and collisions.

**Methods**. Two ramp designs were studied in an experimental setting—a ramp made of plastic poultry slats (grid ramp, GR) and a ramp made of wooden rungs (ladder ramp, LR). Sixty-four young female hens were trained to move to a food reward and this was used to test their behavioural responses when first negotiating the two different ramps during individual tests. Both upward and downward transitions were studied. Ramp preference was also tested using a room that replicated a commercial single-tier system with both types of ramp available. Birds were placed in this room in groups of 16 for three days and their use of the ramps studied.

**Results**. A greater percentage of birds successfully completed (reached the reward bowl) on the GR than the LR during both upward (58% vs 37%) and downward (83% vs 73%) transitions, and a smaller percentage of birds made zero attempts to use the GR than the LR (upwards: 13% vs 56%, downwards: 8% vs 26%). When making a downward transition, more hesitation behaviours were seen (head orientations, stepping on the spot, moving away) for the LR. However, more head orientations were seen for the GR during the upward transition. Birds were more likely to abort attempts (an attempt began when a bird placed both feet on the ramp) to move up the GR than the LR. Birds took longer to negotiate the LR than the GR in both directions, and more pauses were seen during a successful upward transition on the LR. Birds were more likely to move down the GR by walking/running whereas birds tended to jump over the entire LR. More collisions with the food reward bowl were seen for the LR. In the group tests, birds preferred to use the GR, with more transitions seen at all timepoints. However, in these tests, birds preferred to rest on the LR with greater numbers of birds counted on this type of ramp during scan sampling at all timepoints.

**Discussion**. Behavioural results suggest that the GR was easier for the birds to use than the LR, particularly on the downward transition. The GR was also less likely to result in collisions. However, the upward transition may be more difficult on the GR for some birds, potentially because of the inability to pause on a level surface during the transition. The results suggest that the GR was preferred by pullets for moving between a raised area and the ground but the LR was preferred for resting.

Corresponding author
Isabelle C. Pettersson,
i.pettersson@bristol.ac.uk

## INTRODUCTION

Loose-housed systems for laying hens are a growing alternative to cage systems, representing 52% of all egg production in the UK in 2016 (*DEFRA, 2016*). All loose-housed systems incorporate raised tiers that hens must negotiate to access different resources. Single tier (or flat deck) systems consist of a ground level litter area and a raised slatted area where feed, water and nestboxes are usually available. Multi-tier (or aviary) systems also have ground level litter but an additional 2–4 raised tiers, usually stacked on top of each other containing feed, water and nestboxes on different levels. The height of the lowest raised area in all designs varies but is often over 80 cm. When outdoor access is provided (in free-range systems) hens may additionally have to negotiate popholes (openings in the walls of the house that lead to the range) that are frequently raised above floor level.

The ability of hens to negotiate level changes in the house without injury has been questioned, as evidence suggests that collisions are commonplace in loose-house systems (for a review see *Harlander-Matauschek et al., 2015*). Levels of keel bone fracture in flocks can exceed 80% by the end of lay (when birds are removed from commercial systems, usually at 72 weeks) (*Wilkins et al., 2004*; *Wilkins et al., 2011*); however, the addition of ramps to multi-tier systems has been shown to reduce falls and fractures (*Stratmann et al., 2015*). Difficulty traversing level changes may also limit birds' access to important resources and thereby cause frustration.

When the height difference between slats and litter is great, producers often provide some sort of ramp for the birds, although there is no legal requirement to do so and scientific evidence concerning the optimum design, size and placement of such ramps is not currently available. In single-tier systems, the plastic slats used for the raised area are sometimes used to create grid-type ramps along the edge. However, this type of ramp reduces available litter space (as birds cannot access the area underneath the ramps) and so producers may be reluctant to use this design and may therefore provide ladder-like ramps with rungs. In a recent study of 16 free-range flocks, eight flocks were provided with grid-style ramps and six with ladder-style ramps (*Pettersson, Weeks & Nicol, 2017*).

Previous work in commercial flocks observed signs of hesitancy and difficulty (such as pacing, stepping on the spot and repeated crouches) in hens transitioning to the litter in both single and multi-tier systems (*Pettersson, Weeks & Nicol, 2017*). These signs of hesitancy were reduced when full width slatted ramps were present (*Pettersson, Weeks & Nicol, 2017*). Whether ramps should be provided along the entire width of the raised area or intermittently has not been researched, although *Pettersson, Weeks & Nicol (2017)* demonstrated that birds still attempt (and struggle) to descend to the litter using non-ramp areas, even when intermittent ramps are provided. No work has looked at whether there is a preference for different types of ramp or whether hens find a certain type of ramp easier to negotiate. This type of study is more easily conducted under controlled experimental conditions than on a commercial farm, so in the work reported here we compared the

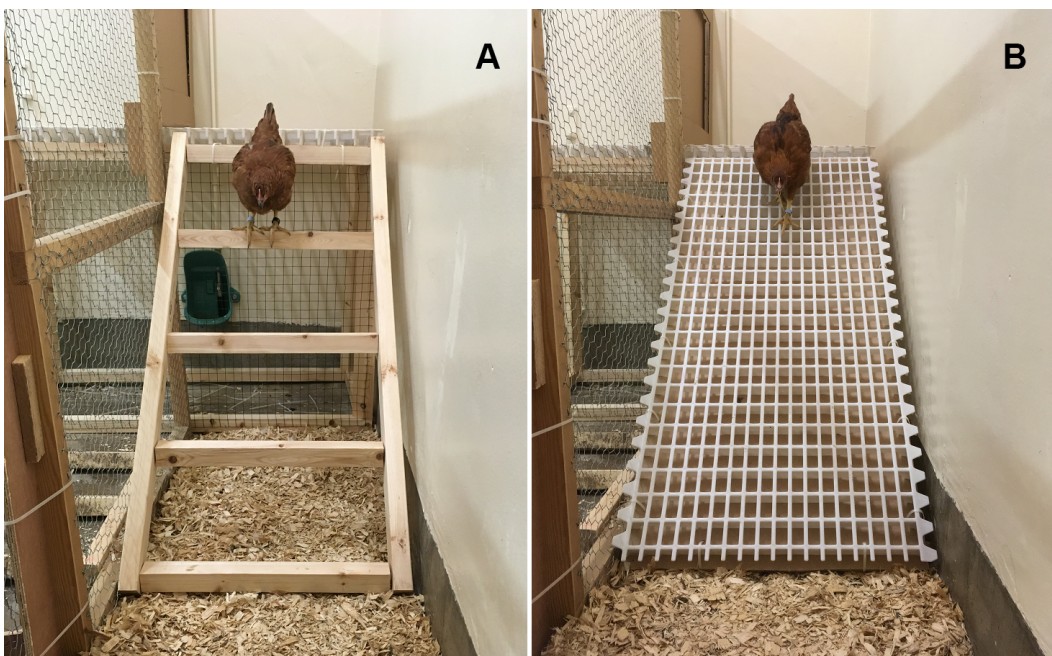

**Figure 1** **Photographs of the two ramps used in this study.** These images are taken of the individual testing ramps although the design was consistent for both individual and group tests. (A) Ladder ramp (LR), (B) Grid ramp (GR). Photographs taken by Kate Norman.

behaviour of hens when negotiating two different types of ramp that are often used in commercial egg laying systems. The main aims of the study, in a small experimental set-up, were to:

  i. Compare for two types of ramp, the behavioural responses of pullets when traversing them to access a food reward.

 ii. Investigate whether hens kept in small groups show a preference for using one ramp type over another.

iii. Determine whether behaviour indicative of difficulty negotiating the ramp and preferences for ramp type differ with direction of movement (upwards or downwards).

## MATERIALS AND METHODS

### Ramp design

The two ramp designs that were studied in this experiment were the "Grid ramp" (GR) and the "Ladder ramp" (LR). See Fig. 1 for photographs. The grid ramps consisted of a 6 mm thick piece of medium-density fibreboard (MDF) covered fully by white plastic poultry slats (Jansen). The ladder ramps were built from planed timber with rungs (4.4 cm × 4.4 cm square) spaced 30 cm apart. These designs were chosen as they are similar to those regularly used by producers on commercial farms.
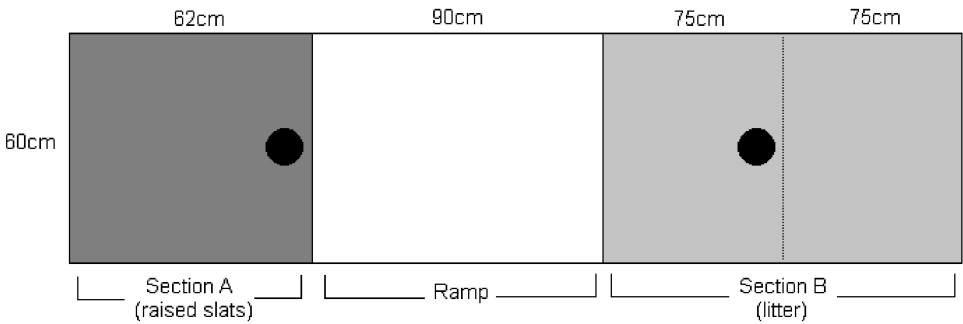

**Figure 2** **Plan of the individual testing pen.** Black circles represent bowl location (at A for upward tests and B for downward tests). The dotted line represents the barrier used only for upward tests.

## Birds and housing

Sixty-four British Blacktail pullets were obtained from an organic commercial rearer at 8 weeks of age. All birds had been given access to small raised tables ($L$:360 cm × $W$:60 cm × $H$:50 cm) from three weeks of age. These had plastic slats on top and two had been provided per flock of 2,000 birds. On arrival, birds were split into four groups, and leg-tagged for identification.

The birds were housed in two identical home rooms, with two groups housed together per room (366 cm × 305 cm). The two home rooms had solid walls and a corridor between so although some bird noise may have been heard between the two, this was minimal. Wood shavings were provided as well as two bell drinkers, two hopper feeders and a single raised table in the centre of the room. This structure had a wooden frame with white plastic poultry slats on top ($L$:120 cm × $W$:60 cm × $H$:50 cm)—matching the structures available at the rearing shed, but shorter in length. In one of the home rooms, a LR was affixed to one long side and a GR to the other (ramp angle: 61°). Both ramps measured $L$:57 cm × $W$:120 cm with only one central rung on the LR (approximately 30 cm from the top and bottom of the ramp). In the other home room, ramps were not provided up to the available structure. Despite two rearing conditions; all birds were used for this study.

Birds were fed ad libitum with layers mash (Farm Gate Feeds, Staffordshire, UK) and the light period was 12 h light, 12 h dark. This project was approved by the University of Bristol's Animal Welfare and Ethical Review Body (UIN: UB/17/046).

## The individual testing room

A long, narrow pen (302 cm × 65 cm) on one side of a separate room was used for individual testing (see Fig. 1). Three sides of the pen were the concrete walls of the room with the remaining side built from a wooden frame and chicken wire so that the hens were visible during testing. See Fig. 2 for a plan of the testing pen.

During the habituation and training periods a white plastic poultry slat was laid on the floor at one end of the test pen (see Fig. 2, section A) with wood shavings covering the other end (section B). During testing, the ground level plastic poultry slat was replaced with slats on top of a wooden structure, 90 cm high. Depending on the test being carried out, either a GR or LR was attached to this raised slatted area using cable ties (ramp angle:

45°). The ramps built for the individual tests were $L$:120 cm $\times$ $W$:57 cm with three central rungs for the LR. During tests where the bird was required to go up the available ramp, a section of the shavings was blocked off using a board (see Fig. 2). At the slatted end of the pen, hens were placed in the pen through a solid cardboard door. At the shavings end of the pen hens were placed in the pen by being lifted over the barrier.

A closed-circuit TV (CCTV) camera (Swann Communications, Port Melbourne, Australia) was installed on the wall at the shavings end of the pen to record behavioural tests. Video quality was 700 TVL.

### The group testing room

A similar room to the home pen (366 cm $\times$ 305 cm) was used for group tests. The far side of the room from the entry door consisted of a raised slatted area ($L$:366 cm $\times$ $W$:120 cm $\times$ $H$:90 cm). This was created using multiple wooden frames with plastic slats on top. The sides of these frames were covered with chicken wire to prevent access under the structures. The rest of the room was covered with wood shavings. A large LR and GR ($L$:120 cm $\times$ $W$:171 cm) were installed side by side to facilitate the transition from raised slatted area to ground level shavings. The LR had three central rungs. These ramps were attached using cable ties to the wooden frame (at 85 cm) and could therefore be moved by the researcher easily as required. Ramp angles were 45 degrees. A small gap of 24 cm between the wall and ladder ramp was always present (in both positions) due to the size of the room. See Fig. 3 for diagram of the group testing room.

A hopper feeder provided food *ad libitum* on the right-hand side of the raised slatted area. A bell drinker was available on the left side of the raised slatted area.

Two CCTV cameras (Swann Communications, Port Melbourne, Australia), one facing each ramp, were installed on the wall above the door. Video quality was 700 TVL.

### Habituation and training

Habituation of the birds took place over a period of 14 days (8–9 weeks of age) with two researchers. Birds were not food-deprived prior to habituation, training or testing. Food was provided ad libitum in the home pens to minimise any effects of hunger and sweetcorn was used as a food reward as birds were generally motivated to access it. Birds were first habituated to human presence and handling concurrently with introduction of the food reward: tinned sweetcorn (days 1–7) in the home room. When birds were calm in the presence of humans and reliably eating sweetcorn from the test bowl, the birds were gradually introduced to the individual test room and pen (days 8–14). The test pen was on one side of the test room. At first, hens were carried into the test room in pairs and fed *ad libitum* sweetcorn from the test bowls. The test bowls were identical: ceramic with a black gloss outside and white gloss inside. This progressed to placement and feeding in pairs in the individual testing pen. Once birds appeared to be comfortable with this, they were habituated to the same process but as individuals. By the end of the habituation period all birds were calm and reliably eating from the bowl when separated from the other birds. Habituation was kept similar for all birds, although particularly nervous individuals were introduced to later steps (e.g., being alone) more gradually.

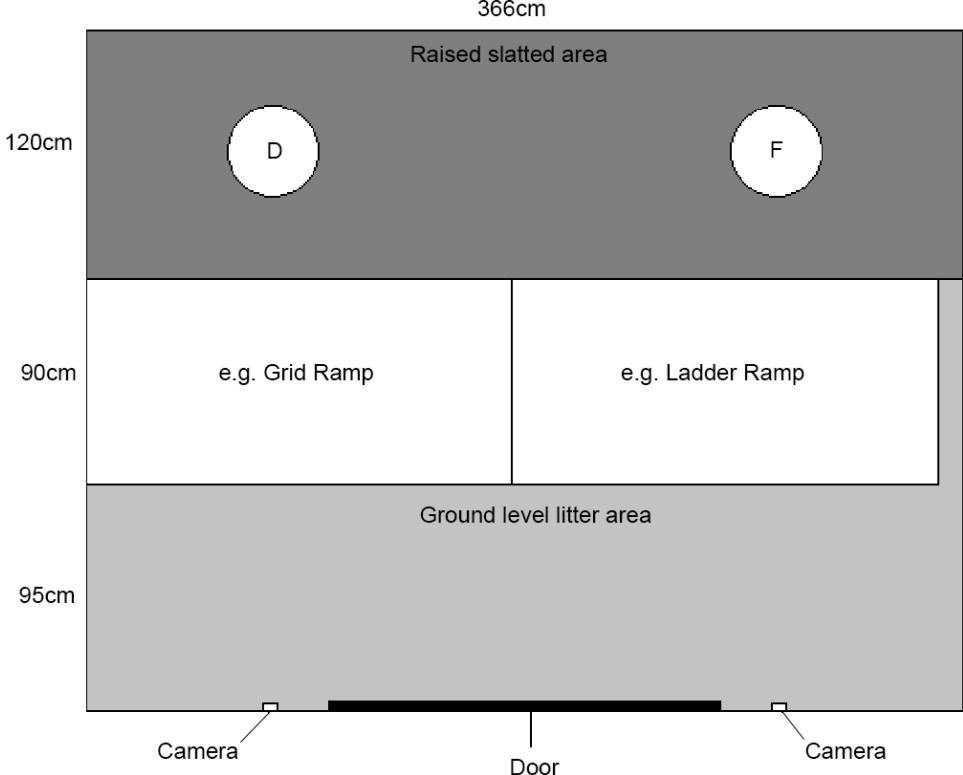

**Figure 3** Plan of the group testing room. D, drinker; F, feeder.

No habituation to the group testing room was performed at any time to avoid any prior experience of the room influencing bird preferences.

Training took place over a period of 14 days (10–11 weeks) following the habituation period. Hens were individually carried into the individual testing pen and placed at one end by one researcher. The bowl containing sweetcorn was already *in situ* at the other end. The second researcher tapped the bowl twice as soon as the bird was placed in the pen (this was new to the birds at the start of training). Once the bird had reached the bowl and eaten, the protocol was repeated for the other direction. The order in which each direction was trained was varied. Each bird experienced this training once a day from days 15 to 25. During the last three days of training (days 26–28) whether the bird succeeded (reached the bowl and ate from it) within 2 min in its two daily tests was recorded (total of six tests). To meet training criteria, the birds needed to be successful in five out of six of these tests. All birds except one (bird 2.10) met this criterion.

### Individual testing protocol

The individual bird testing was undertaken sequentially by group: all of group 1 birds were tested first, then group 2 birds, then group 3 birds, then group 4 birds. As this meant that some birds had a longer break between the training and testing phases a 'recap day' took place for each group before the two individual test days. During the recap day all hens from that group were run through the training protocol in both directions once.

The testing phase took place over two days when the birds were 12–14 weeks. This age was mainly chosen as it was similar to the age when pullets are moved to the laying house commercially (typically 16 weeks, earlier in some organic systems). Each bird experienced four tests, two per day: GR-DOWN, GR-UP, LR-DOWN and LR-UP. The order of testing for each bird was systematically balanced (so that each possible test order occurred within each group) so that testing order could be accounted for. However, birds always experienced the two directional tests for each ramp type consecutively.

For the testing days, the ground level slats in the test pen were replaced with a raised slatted structure and the relevant ramp for the current test was affixed using cable ties (see housing description).

Two researchers were present during the tests and performed the same roles for every bird to avoid bias. An example of the test procedure for an 'upward' test is described below (see Fig. 2):

i. The bowl containing five pieces of sweetcorn was placed at the edge of A so that the bird could see it from ground level.

ii. A barrier was positioned along dotted line dividing section B (in diagram).

iii. Researcher 1 placed the bird at B.

iv. Immediately, researcher 2 tapped the bowl twice using a pencil.

v. The bird was allowed 2 min to reach the bowl. If the bird had not reached the bowl after two minutes she was removed. However, if the bird was actually on the ramp after two minutes, she was given a further minute to complete or to leave the ramp. If she remained stationary after this further minute or left the ramp without reaching the bowl she was removed.

vi. When the bird had eaten from the bowl she was picked up by researcher 2 and removed. The bowl was also removed.

vii. An identical bowl also containing five pieces of sweetcorn was then placed at B. The barrier (denoted by a dotted line in Fig. 2) was removed (as the hens needed more space to land if they jumped).

viii. Researcher 2 placed the bird at A.

ix. Immediately researcher 1 tapped the bowl twice using a pencil.

x. When the bird had eaten from the bowl she was picked up by researcher 1 and removed. The bowl was also removed. See point 'v.' above for the protocol if the bird did not reach the bowl within 2 min.

xi. Her testing was then complete for that day and the bird was returned to the home room until further testing the next day.

## Group testing protocol

After all 16 individuals from a group had completed the individual testing phase those 16 birds were placed the following day in the group testing room. All birds were placed on the raised slatted area between 08:30 and 09:00.

The group testing phase took place over 3 days for each group. This meant that individual testing of the next group could continue while the previous group was in the group testing room. The first day was a habituation day for the birds to familiarise themselves with the

room. No data were recorded on this day. The next two days were considered test days and video footage was recorded between 09:00 and 17:00. To minimise side biases, between 08:30 and 09:00 on both test days all 16 birds were placed into crates and the positions of the two ramps were reversed. The birds were then placed back into the room on the raised slatted area. The position of the ramp was balanced so that two groups started with the GR on the left and two with the GR on the right.

On the fourth day the birds were removed from the group testing room between 08:30 and 09:00 and placed back into their home room. The next group was then placed in the group testing room, once ramps had been changed, to begin its habituation day. This was repeated until all four groups had completed their group testing.

### Behavioural measures recorded
#### Individual tests
Video footage was watched using VLC media player (VideoLAN, Paris, France). Behaviours indicative of hesitation (as used in previous work, *Pettersson, Weeks & Nicol, 2017*): head orientations, crouches, pacing, stepping on the spot and moving away were recorded, as well as whether a bird aborted an attempt (left the ramp before reaching the opposite side) or was successful (reached the opposite side). Latencies of successful transitions and latencies from the start of the test until the bird reached the bowl were measured. Further behaviour was recorded during successful transitions. See Table 1 for a description of the variables recorded from these videos.

#### Group tests
For analysis, observations taken from video footage were split into eight time periods, each of 1 h duration. Attempts were made to identify individuals in these group tests of 16 birds but it proved too difficult. The total number of transitions by birds both up and down each ramp each hour was recorded. A transition began when a bird placed both feet on the ramp (for GR) or a central rung (for LR) and ended when the bird placed both feet on either the raised slatted area or litter area (depending on the direction of movement). If the bird did not complete the transition within 1 min it was not counted as the bird was deemed to be using the ramp for a different purpose other than moving between the litter and slats. The counts were recorded separately for each ramp and for both upward and downward transitions.

Additionally, scan samples were taken of the number of birds on each ramp every 5 min.

### Analysis
All data were analysed using SPSS 23 (IBM, Armonk, NY, USA).

Data on individual birds were analysed separately for upward and downward tests. For variables that did not require birds to complete a transition (e.g., hesitation behaviours), all birds were included, except for one bird that did not meet training criteria and one bird that was tested incorrectly, resulting in $n = 62$ birds for this analysis. The effect of ramp type on the time taken from bird placement in the test runway to reaching the bowl (time to complete the task) was calculated as a percentage of the total time allowed (usually 2 min,

**Table 1  Ethogram for individual testing.** Description of the behaviours recorded during individual bird testing.

| Behaviour name | Downward transition | Upward transition |
|---|---|---|
| Head orientation | The bird lowers its head and neck and looks at the litter. Head orientation towards the walls of the pen and the ramp itself not included. | The bird raises its head and neck and looks at the slats. Head orientation towards the walls of the pen and the ramp itself not included. |
| Crouch | The bird lowers the body while the head is orientated towards the litter. | The bird lowers the body while the head is orientated upwards. |
| Pace | The bird walks along the edge of the slat next to the ramp. A pace must be followed by a head orientation/crouch/step within 10 s. | The bird walks along the edge of the litter next to the ramp. A pace must be followed by a head orientation/crouch/step within 10 s. |
| Step | While facing out towards the ramp, and at the edge of the slats, the bird raises its feet individually and places them back down in a similar location as if adjusting its position. | While facing towards the ramp, and at the edge of the litter, the bird raises its feet individually and places them back down in a similar location as if adjusting its position. |
| Move away | The bird orientates its body over 90° away from the ramp after first showing at least two intention behaviours (e.g., two head orientations or one head orientation and one crouch). | The bird orientates its body over 90° away from the ramp after first showing at least two intention behaviours (e.g., two head orientations or one head orientation and one crouch). |
| Escape attempt | The bird is orientated towards a runway wall and either jumps towards it or crouches as if to take off. | The bird is orientated towards a runway wall and either jumps towards it or crouches as if to take off. |
| Turn | The bird turns the whole body at least 90° while on the ramp. | |
| Pause | The bird stops travelling on the ramp for at least 2 s. | |
| Full jump | The bird jumps all the way from the slats to the litter or vice versa without making contact with the ramp. | |
| Part jump | For grid ramp: The bird jumps partway up or down the ramp but walks/runs the rest (the bird does make contact with the ramp). For ladder: The bird skips at least one rung of the ladder but makes contact with at least one of the central rungs. | |
| No jump | For grid ramp: The bird walks/runs the whole ramp. For ladder: The bird uses all of the central rungs. | |
| Collide with bowl | The bird hits the bowl on landing with any body part. | |
| Stumble/Fall | The bird appears to stumble/trip during the transition but does not fall or the bird falls during the transition, falling onto its side or back. | |

but 3 min for birds still on the ramp at 2 min (this occurred in seven tests), see methods). Birds that did not complete the test were recorded as having a percentage of 100.

For variables that required birds to complete a transition (e.g., time taken to make a full upward or downward transition), only birds that successfully completed the tests for both ramps were included so that individual comparisons could be made. For the upward transition, this resulted in $n = 21$ and for the downward transition $n = 44$. Additionally, some further birds had to be removed for the analysis of certain variables (e.g., number of pauses on the ramp) if they completed the test by jumping over the ramp completely. Variables were analysed using paired $t$-tests or Wilcoxon tests (where the assumptions of the $t$-test could not be met). Square root and log transformations were performed in attempts to use the parametric $t$-test, but achieved normality in only one analysis (latency of a successful attempt, upward transition). Means and standard deviations have been reported throughout this paper as the nature of the data meant that medians were often

the same even where very significant differences in distribution were detected. Means have therefore been used.

A Spearman's rank correlation test was also used to look at a potential relationship between two variables: the percentage that performed a full jump and the percentage that collided with the bowl. For this, data for both ramp types were combined in one analysis.

Some of the data from individual testing could not be analysed on an individual bird basis as the variables were nominal and therefore had to be summarised (e.g., percentages of successful birds). As only a single percentage was produced per variable, per behavioural test formal statistical tests were not performed but graphs were produced for visual inspection.

Similarly, the data from group testing (e.g., number of transitions up and down the two types of ramp; and average numbers of birds on each ramp type) also had a very low sample size ($n = 4$ groups) so formal statistical tests were not performed. Instead, graphs were produced for visual inspection of differences between ramp types and to examine diurnal effects. Initial screening showed that Day had no significant influence on these variables so these data were averaged over the two days for each timepoint. For the graphs, groups were also averaged.

## RESULTS

### Individual tests

Of the 62 birds tested, 36 birds (58%) successfully completed the GR-UP test, 51 birds (82%) completed the GR-DOWN test, 23 birds (37%) completed the LR-UP test and 45 birds (73%) completed the LR-DOWN test.

There were some effects seen on measures of hesitancy. The number of head orientations exhibited by each bird was significantly greater for the GR (1.98, SD:1.17) than the LR (1.50, SD:1.40) when performing an upward transition ($Z = -1.967$, $n = 62$, $p = 0.049$). The opposite was true for the downward transition with significantly greater head orientations when negotiating the LR (2.73, SD:2.73) than the GR (1.47, SD:1.02) ($Z = -3.646$, $n = 62$, $p < 0.001$).

Birds performed more steps on the spot before moving down the LR (0.61, SD:1.08) than the GR (0.10, SD:0.53) ($Z = -3.646$, $n = 62$, $p < 0.001$). There were too few instances of this behaviour (<10 birds per ramp type) to test the upward transition. Pacing behaviour could not be tested for the same reason.

The number of times a bird moved away from the ramp when negotiating a downward transition was significantly affected by ramp type ($Z = -2.702$, $n = 62$, $p = 0.007$) although the number of occurrences per bird were low for both ramp types. On average birds moved away from the GR 0.08 times (SD:0.38) and the LR 0.39 times (SD:0.82). Ramp type did not have a significant effect on this variable for the upward transition.

The number of aborted attempts to use the ramp (where the bird left the ramp before completing the transition) for an upward transition was significantly greater for the GR (0.42, SD:0.64) than the LR (0.08, SD:0.27) ($Z = -3.622$, $n = 62$, $p < 0.001$). The downward transition could not be tested due to insufficient instances of aborted attempts.

Ramp type significantly influenced the percentage of the permitted 2 min (3 min for seven tests) test time used to complete the task for both the upward and downward

transitions. Birds took significantly longer to negotiate the LR than the GR for the upward transition, where the percentage of time used was 81.63% (SD:30.71) and 58.24%, (SD:41.80) respectively ($Z = -4.329$, $n = 62$, $p < 0.001$). The same was true for the downward transition, where the percentage of time used was 34.13% (SD:43.36) for the LR and 25.84% (SD:39.74) for the GR ($Z = -3.247$, $n = 62$, $p = 0.001$).

The latency of a successful upward transition was significantly affected by ramp type ($t(20) = -3.062$, $p = 0.006$). Mean latency on the GR was 8.48 s (SD:9.46) whereas mean latency to move up the LR was about three times longer (25.10 s, SD:33.06). There was no significant effect of ramp type on latency of the downward transition ($Z = -1.397$, $n = 44$, $p = 0.162$).

The number of pauses on the ramp during a successful upward attempt was significantly affected by ramp type ($Z = -2.905$, $n = 21$, $p = 0.004$) with a mean of 0.48 (SD:0.93) on the GR and 1.57 (SD:1.08) on the LR. There was no significant effect of ramp type on number of pauses during the downward transition ($Z = -1.000$, $n = 11$, $p = 0.317$).

The number of turns on the ramp during a successful attempt was not affected by ramp type for either the upward transition ($Z = -1.890$, $n = 21$, $p = 0.059$) or the downward transition ($Z = -1.000$, $n = 11$, $p = 0.317$).

See Fig. 4 for a graph showing the percentage variables for each ramp type. A greater percentage of birds were successful at the downward transition compared with the upward transition for both ramp types. The percentage of birds that successfully completed the tests was higher for the GR with a difference of approximately 20% for the upward transition and 10% for the downward transition. The percentage of birds that made no attempts to complete the ramp during individual testing was higher on the LR, with a greater difference of over 40% for the upward transition. No birds collided with the bowl when performing an upward transition on either ramp, but it was seen in 36% of downward transitions on the LR when compared with 6% on the GR. There were generally few incidences of birds stumbling/falling with no birds doing so on the GR for the upward transition but 4% doing so on the LR. For the downward transition, a greater percentage of birds also stumbled/fell on the LR (9%) compared with the GR (2%). The percentage of birds that jumped fully over the LR (without touching the ramp) during the downward transition was 71% with no birds doing so on the GR. The percentage of birds that performed a full jump during the downward transitions was highly positively correlated with the percentage of birds that collided with the bowl (Spearmans Rho = 0.937, $n = 8$, $p = 0.001$).

Virtually all birds (>90%) used the whole GR and did not jump at all in either direction and no birds jumped fully up either ramp. Partly jumping the ramp (or missing rungs on the LR) was seen with both ramp types. Approximately 8% of birds partly jumped the GR in both directions but it was seen more on the LR than the GR with 52% and 16% of birds partly jumping the upward and downward transitions respectively.

## Group test

At all timepoints, more transitions were seen on the GR than the LR (Fig. 5). The difference is generally greater for the downward transition. Ramp use showed some fluctuation throughout the day with greater use at the start and end of the 8 h observation period.

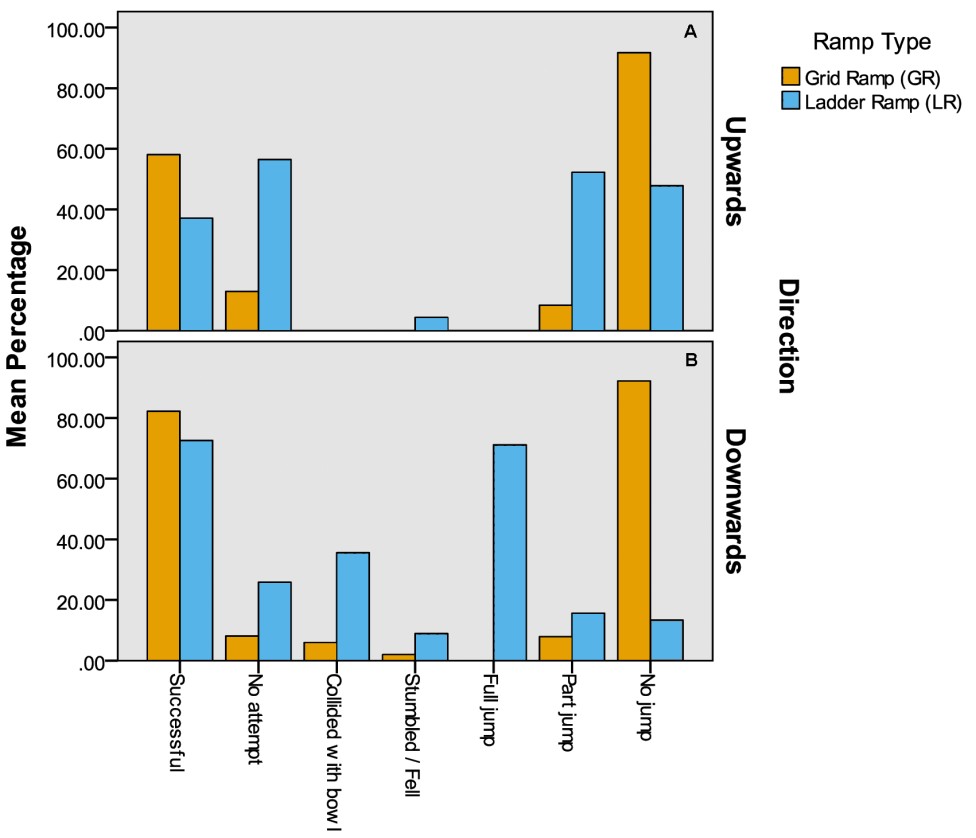

**Figure 4 Individual testing—behaviours.** Descriptive results of bird behaviour when negotiating the ramps. (A) Upward transitions, (B) downward transitions.

For all timepoints, many more birds were seen on the LR than the GR (Fig. 6). Very little fluctuation over time was seen for the GR, but a clear increase in birds on the LR was seen throughout the observation period with a peak between 3 and 4 pm. This then dropped off in the last hour of the observation period.

Birds appeared to prefer using the ramp on the right-hand side of the room but this was fully accounted for by ensuring that every group experienced both ramp types on both sides of the room.

## DISCUSSION

The first aim of this study was to investigate differences in individual bird behaviour and use of the two ramp types. Behaviours indicative of hesitation or difficulty with a level-change have been used in previous studies (*Pettersson, Weeks & Nicol, 2017*; *Lambe, Scott & Hitchcock, 1997*) and include head orientations, crouches, stepping on the spot, pacing and moving away without performing a transition. When attempting a downwards transition, the number of head orientations, steps on the spot and incidences of birds moving away were greater for the LR than the GR, suggesting increased hesitation and difficulty using this ramp design. There were fewer differences in these behaviours between

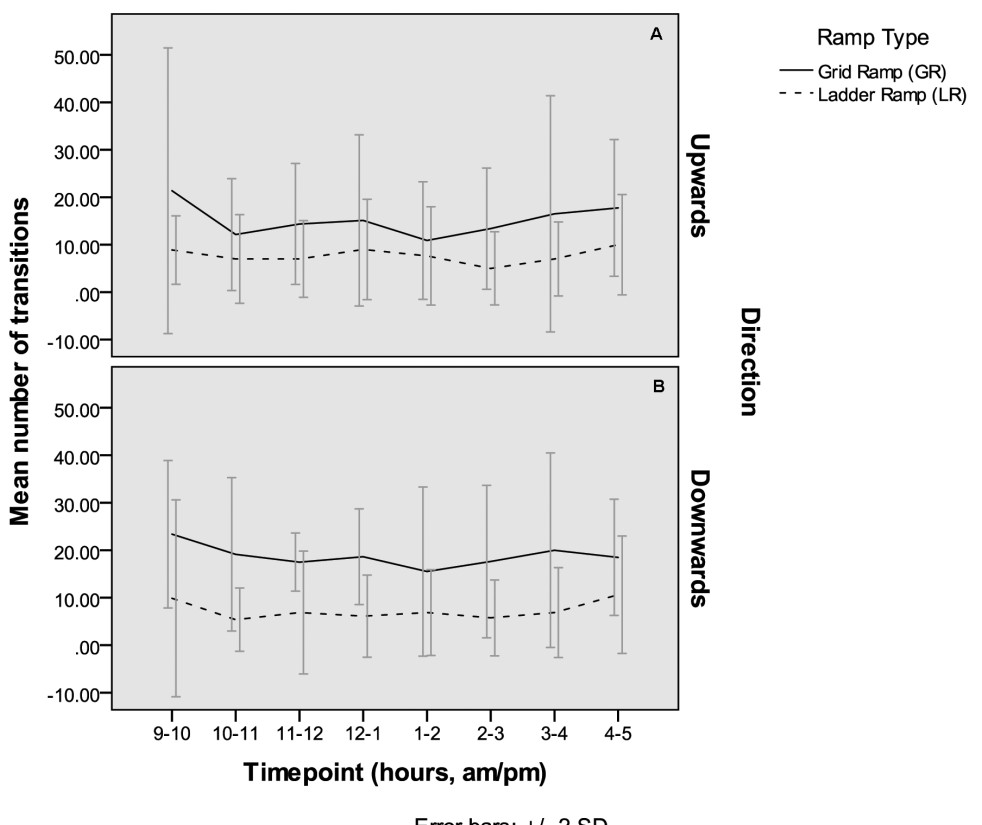

**Figure 5  Group testing—number of transitions.** Hourly mean number of transitions on each ramp type, when both were available, over the two 8 h group test days, averaged across the two days and the four groups. (A) Upward transitions, (B) downward transitions.

the ramp types for the upward transition although the GR elicited more head orientations in this direction. Similarly, more aborted attempts were recorded for the GR than the LR in this direction. A possible explanation for this is that the GR required a bird to walk up a steep incline, rather than jump between rungs as with the LR. When aborting an attempt on the GR birds often stopped midway up, turned and ran back down. Therefore, if a bird paused on the GR they had to stand on an inclined surface, so perhaps could not balance and therefore aborted their attempt. The LR may have allowed the bird to pause without losing balance. Indeed, the average number of pauses during a successful upward attempt was greater for the LR than the GR. Due to the nature of the LR, more pauses might be expected as the bird must jump between rungs. However, to be considered a 'pause' in this study the bird had to stop moving for 2 s and many birds did not pause at each rung on the ladder ramp suggesting that a greater number of pauses on the LR cannot be explained by the ramp design alone.

There were very clear effects of ramp type on how the bird used the ramps. Over 90% of birds fully used the GR, either running or walking on the ramp for both upward and downward transitions. Less than 50% used the LR in this way in both directions. Partly

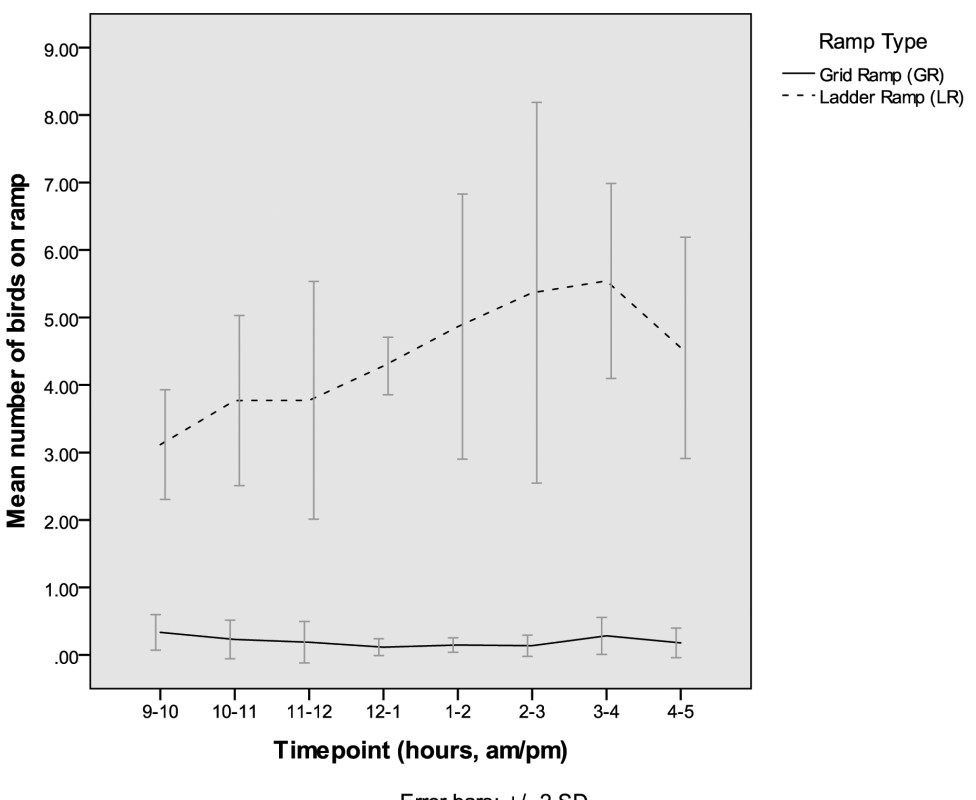

**Figure 6  Group testing—number of birds on ramps.** Mean number of birds on each ramp type (from 5-min scan sampling) over the two 8 h group test days, averaged across the two days and four groups, indicating that birds rested on the ladder ramp but did not rest on the grid ramp (both at 45°).

jumping (taking off midway or skipping rungs) was seen for both types of ramps, but occurred more often on the LR. This was likely because a jump was already required (between rungs) so to skip a rung or two may not have proved to be much more difficult for some birds. Birds were far more likely to fully jump over the LR than the GR when negotiating a downward transition. In fact, no incidences of this were seen for the GR at all but over 70% of all downward transitions for the LR involved a full jump. Some evidence shows that hens are less likely to land poorly on litter than on perches (*Campbell et al., 2016*) so it may be perceived as safer for hens to jump the whole LR instead of attempting each rung. Interestingly, birds were far more likely to collide with the bowl when transitioning down the LR and this was highly correlated with the percentage of birds that jumped fully. This suggests that a jump from the 90 cm high slats to the ground level litter was somewhat uncontrolled, thereby increasing the chance of collisions. Similarly, a greater percentage of stumbles and/or falls were seen for the LR, particularly on the downwards transition. Keel bone fractures are likely caused by collisions in laying hen housing, and systems where chances of falling/jumping from a height are more likely (e.g., where elevated perches are available) see more fractures (*Rodenburg et al., 2008*; *Wilkins et al., 2011*).

A greater percentage of hens successfully completed the GR in both directions but this difference was greater for the upward transition. As already discussed, the birds often jumped fully over the LR when transitioning down, and this may have provided an easier option. When transitioning up the ramp however, such a large jump would be much more difficult (no birds did this) meaning the bird would need to use the ramp itself. Therefore, if birds need to use a ramp, they appear to have more success with a GR. Supporting this, the percentage of birds that did not make any sort of attempt to reach the reward bowl was greater during LR tests, particularly during upward tests. The latency of a successful attempt was significantly greater for the ladder ramp during an upward transition but there was no significant difference for the downward transition. The latency from first being placed in the test pen until reaching the bowl (as a percentage of the total time allowed) was greater for the LR for both upward and downward transitions. These results suggest that birds were faster to start making a successful downward attempt with a GR. This is supported by the results on hesitation behaviours described above.

The second aim was to investigate whether hens in small groups showed a preference for using one type of ramp over the other. The grid ramp (GR) was used consistently more than the ladder ramp (LR) in this group setting, for both upward and downward transitions. The GR was used on average, 2 × more for upward transitions and 2.5 × more for downward transitions with some fluctuation in use throughout the day (Fig. 5). More transitions were generally seen in the first hour, likely because the birds had just been placed back in the room and were therefore moving around to explore. As the birds were free to make their own choices about how to move between the slats and litter the results suggest that the hens in this study showed a preference for using the GR over the LR as a ramp. It is possible that the LR was used less because of blocking by other birds. The scan count results indicate that on average, less than 1 bird was counted on the GR at a given time whereas over four birds were counted on the LR. Obstructions, particularly when birds have to jump (as with the LR) can change bird behaviour and increase risk of collision (*Moinard et al., 2005*). As perching and resting on the LR seemed commonplace, and a 'transition' was only counted if the bird moved from slats to litter (or vice versa) within 1 min, it may be that birds mostly used the LR to perch and not as a direct route to reach the upper or lower area.

This apparent dual-purpose of the LR as a ramp and a perch has important connotations when considering which ramp is most appropriate for a commercial system. If birds are attracted to perch on the LR they will block birds that wish to use the LR as way of accessing the upper/lower area, potentially leading to uncontrolled flights and therefore injury as previously discussed. This may then be affected by existing perching available in the system, making the LR more suitable in certain systems than others. Similarly, the efficacy of the LR as a ramp may be affected by the time of day as the number of birds on the LR increased gradually throughout the day, reaching a peak in the afternoon. Diurnal effects of perching are not clear cut—*Faure & Jones (1982)* found that some strains of birds showed increased perching in the afternoon but *Channing, Hughes & Walker (2001)* found decreased perch use and resting behaviour in the afternoon in multi-tier systems. *Carmichael, Walker & Hughes (1999)* found decreased resting in the afternoon and increased movements from

the litter to perches. These results differ from those presented here where birds appeared to perch more in the afternoon and transitions between the litter and raised area did not show a clear difference between morning and afternoon.

Grid ramps may have generally been preferred and easier to use because they do not require the bird to perform any jumping behaviour. Jumping may be risky for the chicken and harder to execute successfully, particularly where landing areas are small, unknown or partially blocked (*Campbell et al., 2016*; *Moinard et al., 2005*).

### Limitations

The main limitations of this study are rooted in its experimental nature. Although a miniature version of a commercial shed was created for the group testing, groups of 16 birds were small compared with a commercial flock of several thousand, and it is possible that ramp preference may be different in a truly commercial setting with greater stocking densities. Additionally, as only four groups were tested, the sample size was too small for statistical testing. With the individual testing, some birds were highly motivated by the sweetcorn reward and so may have exercised less caution than they would usually when using the ramps. By using a repeated measures design (where individuals were compared against themselves), these individual differences could be somewhat accounted for. The birds used were quite young, and it is possible that their behaviour would change when older. However, young birds were used for three reasons. Firstly, so that we could control their rearing environment, secondly so that the birds experienced the tests close to a time they were likely to first come across ramps in a commercial environment (e.g., 15–16 weeks commercially, these birds at 12–14 weeks) and thirdly because we did not want motivation to nest and egg laying behaviour to influence the results.

## CONCLUSIONS

To conclude, the grid ramp design was preferred by the birds to the ladder ramp design as a method of transitioning between a raised slatted area and ground level litter area. Fewer signs of hesitation were observed when birds negotiated the grid ramp, although some birds appeared to find it harder to move up the grid ramp than down. The ladder ramp had a greater risk of collisions when birds were moving down, probably because they tended to jump rather than walk. Ladder ramps were often used as perches and this may reduce their suitability in a commercial system due to the likelihood of blocking of the ramp by perching birds. Based on these results, we would recommend that producers provide grid-style ramps on their farms where possible.

### Funding

All funding was provided by Noble Foods and Stonegate. There was no additional external funding received for this study.

### Grant Disclosures

The following grant information was disclosed by the authors:
Noble Foods and Stonegate.

### Competing Interests

Christine J. Nicol is an Academic Editor for PeerJ. The authors declare there are no competing interests.

### Author Contributions

- Isabelle C. Pettersson conceived and designed the experiments, performed the experiments, analyzed the data, wrote the paper, prepared figures and/or tables, reviewed drafts of the paper.
- Claire A. Weeks and Christine J. Nicol reviewed drafts of the paper.
- Kate I. Norman conceived and designed the experiments, performed the experiments.

### Animal Ethics

The following information was supplied relating to ethical approvals (i.e., approving body and any reference numbers):
   This project was approved by the University of Bristol's Animal Welfare and Ethical Review Body (UIN: UB/17/046).

### Data Availability

   The raw data has been uploaded as Data S1.

### Supplemental Information

Supplemental information for this article can be found online at http://dx.doi.org/10.7717/peerj.4069#supplemental-information.

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
