# Peer review of "The ability of laying pullets to negotiate two ramp designs as measured by bird preference and behaviour"

_PeerJ, doi:10.7717/peerj.4069_

## Round 0.1 · original submission · Minor Revisions

· Academic Editor

Minor Revisions

A key point to consider in your revision is the comment from the first reviewer about the influence of rearing on your data.

As suggested by the reviewer, it would be desirable to include the data on rearing (currently mentioned in the methods) in the present manuscript. The alternative is to remove reference to the data not included and make a stronger case throughout the manuscript for why you have chosen a particular age.

Reviewer 1 ·

Basic reporting

This article is very well-written and clear to read, the level of English is satisfactory.
The introduction is well-referenced and provides a clear background for the need for the study.
The raw data file is supplied and is clear to read.
Figures are relevant and clear although I found the direction of the x-axis labels difficult to read in Figure 4, I think they should be flipped around if possible. I think Figure 1 is extremely valuable and good to see it included.
Minor comments:
Comments on the abstract are first impressions, some information becomes clear when you read the entire manuscript but these were things that were identified when reading the summary.
Lines 29-30: Definition of attempts?
Line 33: Is there any other way to navigate a LR than jumping it?
Lines 35-36: The LR is a clear potential perching apparatus, therefore it would be expected that birds may rest on it more than the GR.
Line 73: ‘ramps’ not ‘rams’
Line 83: comma after ramp

Experimental design

This experiment is original research that is needed for understanding how birds use ramps that are increasingly used in alternative housing systems yet there is currently little scientific literature on the best ramp designs. The controlled individual testing allows for detailed observations of individual bird responses. The experiment was well-designed with a high standard of habituation/training regimes and controls in place but I had a few queries throughout the methods section for additional clarification (see comments below).
However, my biggest query is why the effects of rearing were not included in the current manuscript (written they are to be included in a separate manuscript). As the current manuscript is not that long, it seems like the effects of rearing would be a great addition to this paper rather than being included separately. Particularly as the birds were tested during the rearing age as 12-14 week pullets.

Comments:
Line 97: More information on why this particular design of the ladder ramp was chosen.
Line 104: I assume birds were divided evenly between rooms/groups. Could birds see/hear each other? It is not clear how the rooms/groups are divided? Mesh? Solid walls?
Line 111: comma after ‘room’
Lines 111-112: Clarify here that all birds were used in the current study despite being reared differently.
Line 131: Company details for the camera system?
Line 149: Clarify if any food deprivation prior to testing and how hunger/lack of hunger was potentially controlled for as a motivation factor.
Line 150: Age of birds during habituation stated here would be useful.
Lines 150-152: Did this occur within the home pen?
Lines: 152-154: Were all birds given the same amount of habituation or did habituation depend on individual bird responses. I imagine some birds were quicker to habituate than others?
Lines 154-157: It is not entirely clear to me the difference between test room and test pen. Where precisely were the birds placed for each of these habituation periods?
Line 160: Delete the word ‘because’
Lines 164-165: Was this pencil-tapping also done during habituation or just started during training?
Lines 165-166: Was training always in the same direction first?
Lines 173-174: I am not clear on what is the difference between the 2 groups. From my understanding, birds were reared in two rooms, with 2 groups per room. Therefore, are you only testing birds from 2 groups within 1 room? Or are you referring to 2 rooms (2 groups per room) here?
Line 180: change to ‘testing order per ramp type’
Lines 231-232: Were these the only behaviours used as indication of hesitation? If so, I think the sentence should read ‘including head orientations, crouches…..etc’. Right now it reads as there were also several other behaviours included as indicative of hesitation and I think all should be listed here as this is a novel approach for behavioural observations.
Lines 242-243: Further clarification as why were these not counted? Were these interpreted failed attempts or birds just using the ramps for resting? Based on your observations, if a bird did not complete a transition within 1 min, it was likely the bird was not using the ramp for transition purposes but was just perching on it (if you can make that assumption from the observations).
Thus is this why the 5-min scan samples were included to look at ramp use for resting/perching?
Line 251: ‘included’ not ‘analysed’
Lines 255-256: Would be useful for the reader to state how many birds these exceptions applied to. Particularly for the 2 or 3 min test if this was only a small number of birds.
Lines 269-270: Further details on how interactions were assessed using t-tests.
Lines 271-272: I don’t understand what two variables are being combined. Further clarification for this sentence.
Line 273: ‘some of the data’

Validity of the findings

I think the data are thoroughly analysed with statistical analyses presented where possible. The group tests just provide visual data due to the low sample size but the visual presentations are still valuable.
The discussion is well written and conclusions drawn from the findings are valid. I have a few comments on the results and discussion as detailed below.

Lines 303-306: Perhaps I missed somewhere earlier any statement about measuring/analysing attempts.
Line 307: But sometimes 3 mins? For how many birds? Any effect of ramp type on whether 3 mins was required?
Lines 317-320: Can you comment on pauses with the LR. It almost seems by design, there are pause breaks inserted as there are rungs compared to the continuous nature of the GR. Would you therefore expect birds to pause on the LR compared to the GR, regardless of whether one is more difficult than the other or not?
Line 357: in individual bird behaviour
Line 359: Lambe et al.
Lines 419-425: I think here it would be more valuable to include some discussion of using the LR as a perch and whether this will have impact on bird movement in the system compared to a GR that is typically not used as a perch. Since the LR appears to have a dual purpose, in a commercial system this may impact bird movement in different areas of the house if it is consistently blocked by birds using it for perching rather than transitioning. Discussing diurnal patterns of perching seems a bit out of place here, but perhaps could be included in relation to what is discussed above and any interplay between perching patterns and transition patterns within a housing system. The LR is potentially confounded by its dual purpose and this may then be affected by available perch space within the system, perhaps this could be discussed.
Lines 426-429: For clarification, could birds in the group testing room go from the slats to the litter without using a ramp? Or did the ramp placement eliminate this possibility (I think by your figure the two ramps had no gaps inbetween)? If birds could bypass the ramp completely, did you look at the numbers of birds that did this? Whether the perching on the LR meant more birds were jumping down to the litter and not using a ramp at all? Would be valuable to discuss whether just ladder ramps in a commercial setting that could be consistently blocked may mean more birds execute jumps that could injure them.

Lines 441-443: another reason why the effects of rearing might be valuable to include in this current manuscript rather than as a separate manuscript.
Lines 448-454: I think the confound of the LR being used a perching space needs to be mentioned here as this is likely to have impact in a commercial system as to what the birds use the ladder for.

Reviewer 2 ·

Basic reporting

Line 33: Collisions with the food reward bowl were “greater.” Specify more frequent instead of greater, as greater collisions implies more significant impact, the forces of which were not measured in this study.

Line 54: define “pop-hole” and/or, if possible, provide a citation.

Line 58: define “end of lay” for the non-specialist in poultry rearing. How long does this interval last, or how old are the birds at the end of lay?

Line 73: typo “ram” should be “ramps”

Line 96: spell out MDF

Line 131: Spell out CCTV. Add sampling rate (frame rate) and video pixel resolution.
Line 180: “Systematically balanced” give slightly more detail. Presumably this means non-random, that every other bird was treated to a different order, or was it a different order for a given group of birds (tested individually within a group). This is relevant to interpreting the confidence intervals for individual performance criteria.

Lines 261-266: It seems that only Z statistics are reported. If that is the case, it would be better to state that non-parametric tests were used because assumptions of parametric tests could not be met, then delete the specific reference to paired t-tests and repeated measures ANOVAs.

Line 266 (and figures): 95% CI’s are reported in the figures. Why not use standard deviations instead? Otherwise, clarify how you calculated the 95% CI’s. It would seem that an N of 4 groups would lead to unnecessarily large CI’s.

Line 286: Add percentages here to the numbers, as percenages are more broadly relevant than the exact numbers

Line 323: I suggest starting with sentence two and putting (“Fig. 4”) at the end of the sentence instead of using the present opening sentence (to eliminate redundancy).

Line 343: As previous, start with second sentence and then direct the reader by putting (“Fig. 5”) as the end of the sentence (delete current topic sentence).

Line 351: Here it is not clear what you mean by “fully accounted for” Expand slightly on detail.

Discussion, General: Two issues would be useful to explore, even if briefly, in the discussion (for example, perhaps near lines 391-394, or 426-427). You do not describe whether the birds ever fell or appeared to lose locomotor control, and you do not mention whether they ever used their wings to assist during climbing or to outright fly. For the research into keel-bone damage, it is worth mentioning if they birds never fell due to loss of control. Likewise, even in general terms, it would be helpful to understand whether wing flapping and flight increases or decreases in relation to ramp design.

Lines 437-438: this is somewhat confusing because you earlier wrote that you could not analyze the data according to a repeated-measures ANOVA. Clarify with additional detail what you mean by “accounted for”.

Figure 1 (legend). It seems that the A figure is a ladder and the B is a grid, but these are reversed in the legend.

Experimental design

Nothing to add.

Validity of the findings

Nothing to add.

Additional comments

Your research is interesting and worthwhile for improving our understanding of ways to design housing systems for layer hens that will lead to reduced frequency of keel bone damage. Your experimental design and analyses are well conceived and rigorous, and I have no major criticisms. I offer minor comments to help improve clarity in selected instances (basicreporting section).

---

## Round 0.2 · accepted · Accept

· Academic Editor

Accept

Thank you for revising your manuscript to meet the reviewers' requirements and I am pleased to say that it is now acceptable for publication in PeerJ.

Reviewer 1 ·

Basic reporting

Article is well written and the authors have made all suggested changes.

Experimental design

The methods section has been clarified as suggested by both reviewers and is now much clearer to read.

Validity of the findings

The discussion is much improved and reads very well.